# Avoidable Mortality Attributable to Anthropogenic Fine Particulate Matter (PM_2.5_) in Australia

**DOI:** 10.3390/ijerph18010254

**Published:** 2020-12-31

**Authors:** Ivan C. Hanigan, Richard A. Broome, Timothy B. Chaston, Martin Cope, Martine Dennekamp, Jane S. Heyworth, Katharine Heathcote, Joshua A. Horsley, Bin Jalaludin, Edward Jegasothy, Fay H. Johnston, Luke D. Knibbs, Gavin Pereira, Sotiris Vardoulakis, Stephen Vander Hoorn, Geoffrey G. Morgan

**Affiliations:** 1University Centre for Rural Health, School of Public Health, The University of Sydney, Sydney, NSW 2006, Australia; timothy.chaston@sydney.edu.au (T.B.C.); katharine.heathcote@sydney.edu.au (K.H.); joshua.horsley@sydney.edu.au (J.A.H.); edward.jegasothy@sydney.edu.au (E.J.); geoffrey.morgan@sydney.edu.au (G.G.M.); 2Health Research Institute, University of Canberra, Canberra, ACT 2617, Australia; 3Centre for Air Pollution Energy and Health Research (CAR), Sydney, NSW 2006, Australia; richard.broome@health.nsw.gov.au (R.A.B.); Martin.Cope@csiro.au (M.C.); Martine.Dennekamp@epa.vic.gov.au (M.D.); jane.heyworth@uwa.edu.au (J.S.H.); b.jalaludin@unsw.edu.au (B.J.); fay.johnston@utas.edu.au (F.H.J.); l.knibbs@uq.edu.au (L.D.K.); gavin.f.pereira@curtin.edu.au (G.P.); Sotiris.Vardoulakis@anu.edu.au (S.V.); stephen.vanderhoorn@research.uwa.edu.au (S.V.H.); 4Health Protection NSW, New South Wales Ministry of Health, St Leonards, NSW 2065, Australia; 5CSIRO, Melbourne, VIC 3195, Australia; 6Environmental Public Health Unit, Environment Protection Authority Victoria, Melbourne, VIC 3001, Australia; 7School of Public Health and Preventive Medicine, Monash University, Melbourne, VIC 3800, Australia; 8School of Population and Global Health, The University of Western Australia, Perth, WA 6907, Australia; 9School of Medicine, Griffith University, Southport, QLD 4222, Australia; 10Ingham Institute for Applied Medical Research, University of New South Wales, Sydney, NSW 2052, Australia; 11Menzies Institute for Medical Research, University of Tasmania, Hobart, TAS 7001, Australia; 12School of Public Health, The University of Queensland, Herston, QLD 4006, Australia; 13School of Public Health, Curtin University, Perth, WA 6102, Australia; 14Telethon Kids Institute, Nedlands, WA 6009, Australia; 15Centre for Fertility and Health (CeFH), Norwegian Institute of Public Health, 0213 Oslo, Norway; 16National Centre for Epidemiology and Population Health, Research School of Population Health, Australian National University, Canberra, ACT 2601, Australia

**Keywords:** anthropogenic air pollution, premature deaths, avoidable mortality, burden of disease

## Abstract

Ambient fine particulate matter <2.5 µm (PM_2.5_) air pollution increases premature mortality globally. Some PM_2.5_ is natural, but anthropogenic PM_2.5_ is comparatively avoidable. We determined the impact of long-term exposures to the anthropogenic PM component on mortality in Australia. PM_2.5_-attributable deaths were calculated for all Australian Statistical Area 2 (SA2; n = 2310) regions. All-cause death rates from Australian mortality and population databases were combined with annual anthropogenic PM_2.5_ exposures for the years 2006–2016. Relative risk estimates were derived from the literature. Population-weighted average PM_2.5_ concentrations were estimated in each SA2 using a satellite and land use regression model for Australia. PM_2.5_-attributable mortality was calculated using a health-impact assessment methodology with life tables and all-cause death rates. The changes in life expectancy (LE) from birth, years of life lost (YLL), and economic cost of lost life years were calculated using the 2019 value of a statistical life. Nationally, long-term population-weighted average total and anthropogenic PM_2.5_ concentrations were 6.5 µg/m^3^ (min 1.2–max 14.2) and 3.2 µg/m^3^ (min 0–max 9.5), respectively. Annually, anthropogenic PM_2.5_-pollution is associated with 2616 (95% confidence intervals 1712, 3455) deaths, corresponding to a 0.2-year (95% CI 0.14, 0.28) reduction in LE for children aged 0–4 years, 38,962 (95%CI 25,391, 51,669) YLL and an average annual economic burden of $6.2 billion (95%CI $4.0 billion, $8.1 billion). We conclude that the anthropogenic PM_2.5_-related costs of mortality in Australia are higher than community standards should allow, and reductions in emissions are recommended to achieve avoidable mortality.

## 1. Introduction

Long-term exposure to ambient air pollution is an established risk factor for a range of cardiovascular and respiratory diseases, contributing to premature mortality and reductions in life expectancy (LE) [1]. Demonstrated nonfatal health associations of air pollution include increased rates of hospitalisation [2], birth defects [3], impaired cognitive function [4], and increased medication usage [5].

Air pollutants that affect health include fine particulate matter (PM) < 2.5 μm (PM_2.5_) in aerodynamic diameter [1]. PM_2.5_ can be emitted from combustion or other processes (primary PM_2.5_) or can be produced via chemical reactions of precursor emissions (secondary PM_2.5_). As an ambient exposure, PM_2.5_ can reach indoor environments and, under certain atmospheric conditions, can travel long distances over several days [6]. Current evidence suggests that there is no safe lower threshold of exposure to PM_2.5_ for mortality because the exposure–response relationship is approximated by a linear function even at very low concentrations [7]. Nonetheless, this remains a strong assumption due to the lack of knowledge about the shape of the exposure–response association at these lower levels [8].

Natural PM_2.5_ includes wind-blown dust, sea salt, organic aerosol from biogenic sources, and emissions from volcanoes and landscape fires. Given the spatial heterogeneity of PM from these sources and the influence of rainfall and wind on their local concentrations, natural PM_2.5_ concentrations vary widely between locations and over time. Anthropogenic sources of PM_2.5_ are responsible for substantial human exposure, emanating from transport and industrial processes such as mining and power generation. Residential wood heaters are also a major source of PM, for example they accounted for 19% of anthropogenic PM_2.5_ emissions and 24% of PM_2.5_ concentrations in Sydney, Australia, during 2010 and 2011 [9]. Although these sources of PM_2.5_ might be expected to increase with population growth and increased economic activity [10], air-pollution control policies have effectively reduced PM_2.5_ concentrations in some high-income countries, as shown by Carnell et al. in the United Kingdom [11]. Several studies show health benefits of reducing anthropogenic PM_2.5_, as reviewed by Rich [12].

Only one study has related all-cause mortality to long-term exposure to PM_2.5_ in the general population in Australia [7], although it was not possible to distinguish between risks from anthropogenic and natural PM sources. Health-impact assessments quantifying the effects on mortality of PM_2.5_ from shipping emissions [13] as well as from wood heaters, traffic, and industrial activities [9] have demonstrated that years of life lost (YLL) and LE offer sensitive indicators of health burdens in Australia. These statistics can also be used to calculate economic costs [14].

Given the ubiquitous but modifiable nature of exposures to anthropogenic PM_2.5_, robust estimates of human-health impacts could be used to inform air-pollution control policies. Herein, we quantified the effect of current levels of anthropogenic PM_2.5_ on mortality in Australia in terms of PM_2.5_-attributable mortality, changes in LE for children, and the economic costs of the associated YLL.

## 2. Materials and Methods

### 2.1. Study Region and Period

We calculated the effect of anthropogenic PM_2.5_ on mortality in all 2310 Statistical Area level 2 (SA2) geographical areas in Australia with age-specific population counts from the Australian Bureau of Statistics (ABS) dataset “Population by Age and Sex, Regions of Australia, Estimated Residential Population 2006–2016” from ABS-TableBuilder (cat. no. 3235.0). We then aggregated attributable numbers of premature deaths, YLL and changes of LE for the entire population of Australia. We chose to start the study period in 2006 to coincide with that census year. PM_2.5_ levels were relatively stable in most states and territories except NT and QLD for the study period 2006–2016 (Figure 1).

### 2.2. Assessment of Anthropogenic vs. Non-Anthropogenic PM_2.5_ Concentrations

Annual average PM_2.5_ concentrations were obtained from a validated satellite-based land-use regression (LUR) model, as described by Knibbs et al. [15]. The regression model uses satellite imagery, chemical-transport model (CTM) simulations and land-use data as predictors, and incorporates direct PM_2.5_ measurements from ambient-air monitoring agencies in Australia [15]. The data are available on request from the Australian Centre for Air pollution, energy and health Research (CAR) https://cloudstor.aarnet.edu.au/plus/f/2454567279. The model was estimated for each mesh-block (MB), which is the smallest area in the Census geography (Figure 2). It is not possible to show MBs in Figure 2 due to the difference in spatial scale. Instead, we have added a small area map to the Appendix A) to demonstrate the small sizes of MB regions, which enable high spatial resolution in our exposure assessment. Anthropogenic PM_2.5_ was defined as the difference between estimated PM_2.5_ concentrations and the 5th percentile of concentrations for all MBs in each state/territory per year. This definition accommodates differences in natural background concentrations between states/territories due to localised influences such as bushfire, dust, and sea salt in the diverse landscapes across the country. This approach is consistent with that taken by the global burden of disease study for estimating the counterfactual level of PM_2.5_ [1].

To validate our state/territory estimates of natural PM_2.5_, we performed sensitivity analyses using 5th percentile PM_2.5_ concentrations for all MBs in Bureau of Meteorology climate zones (Appendix A). This approach gave similar results.

### 2.3. Health Outcomes

Mortality data for years 2006–2016 by age and state/territory and corresponding population data were accessed from the Australian Bureau of Statistics (Cat. No. 3302.0—Deaths, Australia, available from the ABS.Stat website: http://stat.data.abs.gov.au). Further information about these data sources is presented in the Appendix A. No ethics approvals were needed because we used aggregated data from the public domain.

Age-specific death rates for each state/territory and year were linked with the age-specific populations by year within 2016 ABS SA2 geographical boundaries to calculate baseline mortality levels in each subpopulation. To smooth excess variability in annual deaths, we used three-year rolling average annual age-specific rates.

### 2.4. Quantification of Mortality Attributable to Anthropogenic PM_2.5_

Due to the limited number of Australian epidemiological studies of long term PM_2.5_ air pollution exposure and mortality, we used a relative risk (RR) function estimated from a meta-analysis of European and North American studies [16], as recommended by the World Health Organization (WHO) [17]. A pooled RR of 1.062 (95% CI 1.041, 1.084) per 10-µg/m^3^ increments in long term annual average PM_2.5_ exposures of people aged ≥30 years is recommended for health-impact assessments of PM_2.5_ [16,17]. The reviewed studies were performed in countries with similar levels of economic development, similar demographic characteristics, and similar patterns of mortality as those in Australia, albeit with higher air-pollution concentrations [16]. We used this RR to estimate the attributable numbers (AN) of deaths caused by long-term PM_2.5_ exposure for each SA2. We calculated AN based on estimates of local anthropogenic PM_2.5_ and then aggregated to a national total using the following equation:(1)$$AN=\sum^{\;}\left(1-e^{\left(-\beta \Delta X_{ij}\right)}\right)\;\times Expected_{ij}$$
where *Expected_ij_* is the death count estimated by applying mortality rate in age-group *i* by age-specific population counts in SA2 2016 census area *j*, *β* = log (RR)/10 and Δ*X_ij_* is the annual anthropogenic PM_2.5_ concentration in SA2 *j*.

### 2.5. Life Expectancy Calculations

Life tables were generated for each year in each SA2, and LE at birth was calculated for 5-year age groups up to age 85-plus. To quantitatively assess the health impact of anthropogenic PM_2.5_, LE for a hypothetical counterfactual population without anthropogenic PM_2.5_ was calculated by subtracting PM_2.5_-attributable numbers of deaths in each age group from expected numbers of deaths, as described by Miller and Hurley [18]. See Appendix A for more details (Appendix A).

### 2.6. Economic Valuation

To determine the economic value of removing all anthropogenic PM_2.5_ in Australia, we discounted the 2019 willingness-to-pay value of a statistical life year (VSLY = $213,000) by 3% annually [19] and summed for each of the remaining potential life years in each age group. The resulting age-specific value of statistical life (VSL) estimates were then multiplied by corresponding attributable numbers of deaths (averaged for the years 2006–2016) and were summed across all age groups. For more details see Appendix A.

Data preparation and analyses were performed using the R language and environment for statistical computing (version 3.4.4, R Core Team Vienna, R Foundation for Statistical Computing, Vienna, Austria) and MS Excel (Microsoft, Redmond, Washington, DC, USA).

## 3. Results

### 3.1. Exposure Assessment

Nationally, the long-term population weighted average PM_2.5_ concentration across the years 2006–2016 was 6.5 µg/m^3^ (MB min 1.2–max 14.2), and the anthropogenic component was 3.2 µg/m^3^ (MB min 0–max 9.5). Figure 1 shows the modelled 2015 estimates of the annual average PM_2.5_ in ABS MBs across the country and in the major cities. PM_2.5_ concentrations clearly vary across Australia, reflecting the various natural and anthropogenic contributors to ambient PM, such as dust, sea salt, bushfire smoke, and emissions from transport, industry, agriculture, and residential wood heaters.

Table 1 shows estimated average anthropogenic, non-anthropogenic and total PM_2.5_ concentrations in Australia based on the 5th percentile MB level of each state/territory for each year. Average total PM_2.5_ for the entire country varied little between years (Table 1) but differed markedly between states (Figure 2), reflecting diverse ecological conditions across the country. In Queensland and the Northern Territory, PM_2.5_ estimates varied considerably over the study period, due to droughts, floods, and landscape fires (dust storms, bushfire, and controlled burns). In contrast, population-weighted anthropogenic PM_2.5_ concentrations varied little between years, indicating similar pollution sources in Australian capital cities, where most people live.

### 3.2. Mortality Burden

We estimate an average annual mortality burden of 38,962 (95%CI 25,391, 51,669) YLL among people aged 30+ years attributed to anthropogenic PM_2.5_ pollution in Australia between 2006–2016. This is approximately 2% of all mortality or 2616 (95%CI 1712, 3455) deaths. In Table 2, we present annual average mortality burdens in each Australian state and territory. These analyses show that more than 80% of premature deaths occurred in the more populous eastern states New South Wales (NSW), Australian Capital Territory (ACT), Victoria (VIC), and Queensland (QLD).

Based on ANs among Australians of 30+ years-of-age, we estimate that LE among children <5 years-of-age was reduced by 76 (95%CI 50, 101) days due to anthropogenic PM_2.5_ (assuming lifelong exposures). Using the 2019 VSLY of $213,000 with an annual social discount rate of 3% [19], we calculated the value of a statistical life (VSL) for each age group based on remaining LE and estimated an average annual mortality-related cost of anthropogenic PM_2.5_ of $6.2 billion nationally (95%CI $4.0 billion, $8.1 billion).

## 4. Discussion

In this study, we estimate that the mortality burden of anthropogenic emissions of PM_2.5_ in Australia was 2616 excess deaths per year on average (approximately 2% of total mortality), and 38,962 YLL were attributable. In an Australian study from 2016 [13], PM_2.5_ from shipping activities, which use low-quality diesel fuel, were responsible for the loss of 220 years of life among people who died in 2010/11 in the greater metropolitan area (GMR) of Sydney. In a more recent study, 1.2% of all-cause mortality in the greater Sydney metropolitan area was attributed to PM_2.5_ from all anthropogenic sources, corresponding with 5900 YLL annually [9]. In the same study, PM_2.5_ concentrations were estimated using a chemical-transport model (CTM) of eight anthropogenic sources; the total population weighted PM_2.5_ concentration was 5.5 µg/m^3^ with an anthropogenic component of 2.1 µg/m^3^. Our estimates for Sydney are consistent with this anthropogenic proportion, and our estimates of YLL were comparable (data not shown).

Our estimated annual mortality-related cost of anthropogenic PM_2.5_ in Australia was $6.2 billion (95%CI $4.0 billion, $8.1 billion) nationally in 2019 dollars. This is supported by a recent estimate for the special report of the MJA-Lancet Countdown [20], which found that urban PM_2.5_ costs equated to $5.3 billion in 2015 dollars. This is similar to our estimate after adjusting for inflation; however, our methods for exposure assessment and economic valuation with discounting were different. Therefore, comparisons of estimated health-cost estimates should be made with caution. Despite the relatively low levels of air pollution in Australia, the substantial health burden is of public health concern, both in societal and economic terms.

Increased anthropogenic emissions have been associated with increased industrial and economic activities [10], suggesting that concentrations of many pollutants will increase globally over the coming decades without substantial decreases in fossil fuel and biomass combustion [21]. However, economic development has been decoupled from increasing anthropogenic air-pollution emissions in some countries [22] where clean-air policies have been implemented [23]. Moreover, we found no increases in anthropogenic PM_2.5_ over the period 2006–2016 in Australia.

Lelieveld, et al. [8] assessed global PM_2.5_ concentrations and found global mean LE would increase by 1.7 (1.4–2.0) years if all potentially controllable anthropogenic emissions were removed. They estimated total lost LE from air pollution was 2.9 years, exceeded that of smoking (2.2 years of lost LE) [8]. Our estimated loss of life expectancy of 76 days (0.2 years) is similar to that found in [8] for Australia/Oceania combined, but is lower than the global average estimate (1.7 years), due to the lower exposure levels and related mortality rates in Australia. In another worldwide study by the Global Burden of Disease (GBD) 2019 project [24], 1625 (95%UI 508, 2877) deaths were attributed to ambient particulate air pollution in Australia in 2016, whereas our estimate was 2616 (95%CI 1712, 3455) for the period 2006–2016 (data available from Institute for Health Metrics and Evaluation (IHME) website http://ghdx.healthdata.org/gbd-results-tool). Our estimate is 60% higher than that from the GBD study. This difference can be explained by differing datasets used for exposure and death rates, and different exposure–response risk functions and counterfactual exposure.

For context, in a study of Australian smokers in NSW, individual LE of heavy smokers was reduced by 10 years [25]. Given the widespread exposure to anthropogenic PM_2.5_, compared with that of heavy smoking, the population impact may be substantial. For example, the GBD report from 2020 ranked air pollution as the 4th highest risk factor for mortality, with 6.67 million attributable deaths during the period 1990–2019 [24].

We used high-resolution air-pollution models that were informed by monitor data, land-use data and satellite imaging. However, among limitations of this study, we did not analyse regions within states because mortality rates for small areas were not publicly available. Moreover, PM_2.5_ is associated with a broad range of health effects, such as low birth weight and respiratory illnesses, that increase hospitalisation and general-practice visits. The costs of these are not captured by our VSLY estimates, thus only calculating the mortality burden will underestimate the overall impacts of PM_2.5_ pollution on public health and health services. The present study is also limited by the absence of locally derived RR with only one cohort study published [7], and so we applied the RR from a meta-analysis as recommended by the WHO “Health risks of air pollution in Europe—HRAPIE project” recommendations [17]. However, a new meta-analysis has recently been published that found support for a higher RR (1.08), which we have included as a sensitivity analysis [26]. As expected, this showed an increase to our estimated health burden and costs, and supported our conclusion that the burden is substantial. In addition, the global exposure mortality model (GEMM), a non-linear exposure–response function that employs a low minimum-risk threshold was used by Lelieveld, et al. [8]. However, the two meta-analyses [17,26] support our application of a linear RR in the Australian context. A further limitation to our study was the lack of estimates of natural PM_2.5_. We approximated these using the 5th percentile threshold, which is likely to vary less than in reality, but probably overestimates baseline PM_2.5_. This limitation will only be addressed when natural PM_2.5_ estimates from a suitable model are available for the entire country.

## 5. Conclusions

Our findings present some clear implications for policymakers. The estimated burden of premature death attributable to anthropogenic PM_2.5_ shows that this environmental risk factor has a significant impact on public health in Australia, and the health benefits of exposure reductions have been demonstrated in multiple studies. Although ambient annual average PM_2.5_ concentrations have remained relatively stable in major cities in Australia over the past several years, the exposure level is increasing due to increases in the population. Hence, given that the PM_2.5_ exposure–response relationship appears to be linear at the low levels found in Australian cities, PM_2.5_ reporting standards that prioritise continual reductions in PM_2.5_ pollution are urgently required. These are likely to require reductions in emissions of primary PM_2.5_ and of secondary PM_2.5_ precursors across multiple sectors (road transport, domestic heating, industry, and agriculture). Health impact assessments such as this can inform decision-making for urban developments, the energy system, and future studies that assess the costs and benefits of anthropogenic air pollution.

## Figures and Tables

**Figure 1 ijerph-18-00254-f001:**
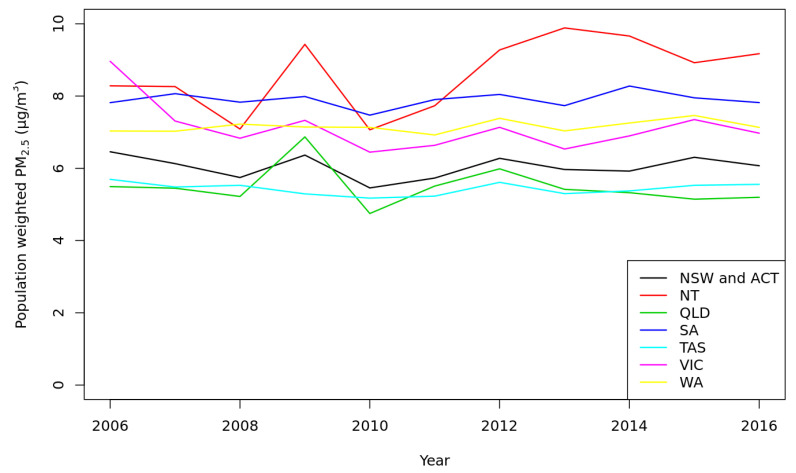
Time-series of population weighted ambient particulate matter (PM_2.5_) concentrations (µg/m^3^) across the years 2006–2016 for all Australian states—NSW, New South Wales; ACT, Australian Capital Territory; NT, Northern Territory; QLD, Queensland; SA, South Australia; TAS, Tasmania; VIC, Victoria; WA, Western Australia.

**Figure 2 ijerph-18-00254-f002:**
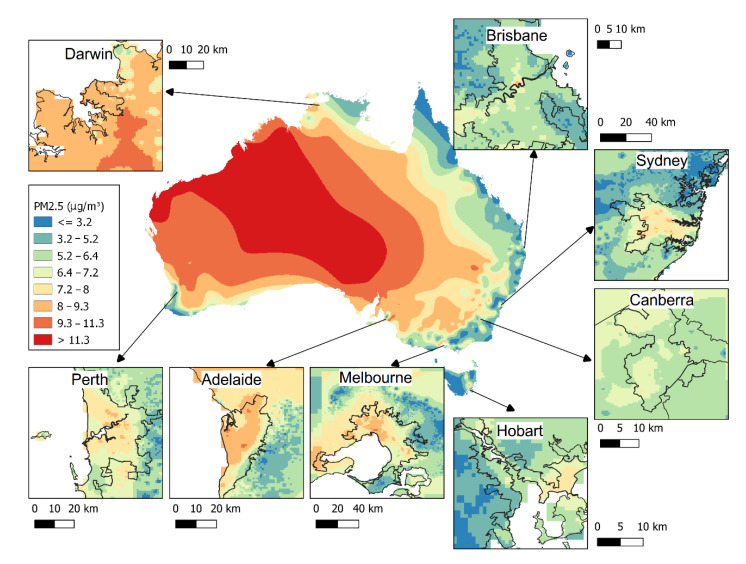
Map of Australia showing modelled estimates of annual average PM_2.5_ (µg/m^3^) concentrations in 2015.

**Table 1 ijerph-18-00254-t001:** Estimated population-weighted national anthropogenic, non-anthropogenic and total PM_2.5_ (µg/m^3^) by year.

Year	Anthropogenic PM_2.5_	Natural PM_2.5_	Total PM_2.5_
2006	3.2	3.9	7.1
2007	3.1	3.5	6.6
2008	3.2	3.1	6.3
2009	3.2	3.7	6.9
2010	3.1	2.8	5.9
2011	3.1	3.1	6.2
2012	3.2	3.6	6.7
2013	3.2	3.1	6.3
2014	3.2	3.2	6.4
2015	3.2	3.4	6.6
2016	3.1	3.3	6.4

**Table 2 ijerph-18-00254-t002:** Average annual mortality burden; Attributable Number (AN) of premature deaths and Years of Life Lost (YLL) in each Australian state and territory; NSW, New South Wales; ACT, Australian Capital Territory; VIC, Victoria’ QLD, Queensland; WA, Western Australia; SA, South Australia; NT, Northern Territory.

Region	AN (95%CI)	YLL (95%CI)
NSW and ACT	940 (615–1241)	13,956 (9094–18509)
VIC	650 (425–858)	9366 (6103–12421)
QLD	517 (338–682)	7925 (5165–10509)
WA	197 (129–260)	3178 (2072–4213)
SA	188 (123–249)	2653 (1729–3518)
TAS	102 (67–135)	1419 (925–1882)
NT	21 (14–28)	464 (303–615)
National	2616 (1712–3455)	38,962 (25391–51669)

## Data Availability

Restrictions apply to the availability of these data. Data was obtained from Dr Luke D. Knibbs and are available from the Australian Centre for Air pollution, energy and health Research (CAR) at https://cloudstor.aarnet.edu.au/plus/f/2454567279 with the permission of Dr Luke D. Knibbs.

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
