# Peer review of "Avoidable Mortality Attributable to Anthropogenic Fine Particulate Matter (PM2.5) in Australia"

_ijerph, 2020, doi:10.3390/ijerph18010254_

Round 1

Reviewer 1 Report

Please see the attached file. Thanks.

Author Response

The reviewer has questioned 1) the definition of our PM2.5 concentration estimate / description in the supplemental document and 2) the research method. Our main response is that we have used the health impact assessment approach that has been widely accepted for air pollution impact studies such as the well-known Global Burden of Disease study (GBD) published in the Lancet [1,2].

Re 1) "describe how to estimate anthropogenic PM2.5 concentrations, put them in the supplemental document".

We did describe in the methods and supplement how we used the 5th percentile of concentrations for all Mesh Blocks (MBs) in each state/territory per year to define counterfactual PM2.5 (the non-anthropogenic level). This approach is consistent with that taken by the GBD study published in the Lancet [1,2]. The sources of our data can be found in the original supplementary information document. To address this comment, we have now also included this information in the main text and cite the source and data access on Lines 102-6.

Re 2) “cannot  know  what  is  the  location  for the subjects from  mortality  data… …the research method has a big problem”.

We respectfully disagree with the reviewer that the method is problematic. We agree with the Academic Editor’s comments that the more direct classic cohort study approach has limitations including bias, can be difficult to generalise and are expensive for long term follow up.  The health impact assessment approach that we employed has been widely used in air pollution impact studies, including its application in the GBD studies[1,2]. We discuss the limitations of the approach but contend it is the most appropriate in the context of the current study's aims.

References:

[1] Abbafati, C., Machado, D. B., Cislaghi, B. et al., Global burden of 87 risk factors in 204 countries and territories, 1990–2019: a systematic analysis for the Global Burden of Disease Study 2019, Lancet, 2020; 396: 10258.

[2] Cohen AJ, Brauer M, Burnett R, et al., Estimates and 25-year trends of the global burden of disease attributable to ambient air pollution: an analysis of data from the Global Burden of Diseases Study 2015, Lancet 2015; 389(10082): 1907–1918.

Reviewer 2 Report

In this paper, authors describe an assessment of the burden of PM2.5 in terms of all-cause mortality in Australia, over the period 2006-2016. The economic burden was also assessed.

I found the paper nicely written, with a thorough description of methods that were used to produce the estimates. However, I think that it could benefit from some improvements.

  • In the section “Assessment of anthropogenic vs. non-anthropogenic PM2.5 concentrations” authors state that they carried out the assessment at a mesh-block kevel, and that such mesh-block can be visualized in Figure 2. However, in Figure 2 the extension of each mesh-block is unclear. I would suggest to better clarify how the territory was divided on the map.
  • The results section about the mortality burden is very concise, and it gives only an average estimate of the annual burden of mortality. I would suggest to enhance it with a description of temporal and spatial trends in the estimated annual burden, as the thorough exposure assessment that has been carried out has the potential to give such spatial and temporal information, that might be very valuable for policy makers. Especially because in the conclusions authors state that “Although ambient annual average PM2.5 concentrations have remained relatively stable in major cities in Australia over the past several years, the associated health burden is increasing due to increases in the exposed population” (lines 243-245). For example, it would be interesting to see how the burden has evolved in all the states that are listed in Figure 1.
  • In the discussion, I would suggest to compare the results with those obtained by the Global Burden of Disease Study, that has recently issued results regarding the period 1990-2019 for the whole World.Such comparison might be interesting

Author Response

Thank you for these suggestions. 

Re 1) “clarify how the territory was divided on the map”.

We have considered the proposed change to Figure 2 to show mesh-block boundaries, but because mesh-blocks are very small areas it is not possible to show them on either a national map or on city-specific inserts. To address the comment, we have added a map to the supplementary information document (Figure S1). This map shows average PM2.5 in 2015 across the country with inset maps of the Sydney region and an inset of the small case study region in Western Sydney. The Western Sydney inset map shows mesh-blocks and demonstrates the high spatial resolution of our exposure assessment. We now also refer to this figure in the main text on Lines 107-110.

Re 2) “description of temporal and spatial trends in the estimated annual burden”.

We now present temporal and spatial trends of the estimated mortality burden as follows: 

  1. We have added spatial trend information in a new Table 2 (Line 194), which shows Attributable Number (AN) and Years of Life Lost (YLL) by each state and territory of Australia.
  2. The US EPA and the UK Committee on the Medical Effects of Air Pollution have considered this issue in detail and recommended that, when assessing the impact of changing exposure, only 30% of the long-term effect estimates should be assumed in the first year after the change. Therefore, reporting year-to-year variation may give a misleading account of the health effects of PM2.5. To recognise the reviewer’s point about the conclusions, however, we have revised the noted sentence for accuracy as follows: “Although ambient annual average PM2.5 concentrations have remained relatively stable in major cities in Australia over the past several years, associatedhealth burden numbers of exposed individuals have increased with population growth” (Lines 261-2). 

Re 3) “compare the results with those obtained by the Global Burden of Disease Study, that has recently issued results regarding the period 1990-2019 for the whole”.

Thank you. We have added this comparison into the discussion, Lines 233-5, and inserted the new citation (25) for the latest GBD report.

Round 2

Reviewer 1 Report

My recommendation is to accept it.

Author Response

Thank you.

Reviewer 2 Report

I thank the authors for addressing my concerns: I think they have properly answered to my first and second requests.

As for the third “In the discussion, I would suggest to compare the results with those obtained by the Global Burden of Disease Study, that has recently issued results regarding the period 1990-2019 for the whole World”, maybe I was not sufficiently clear, therefore I apologize. My suggestion was to compare the results with those of GBD regarding Australia only, and to discuss potential differences and similarities, as authors nicely did for Lelieveld et al. publication, described in lines 225-230.

Author Response

Thank you for this suggestion. We have now included on lines 230-236 a comparison of the results from the latest GBD 2019 estimate for Australia (1625 attributable deaths) compared to our estimate (2616 attributable deaths). We discuss the difference showing our estimate is higher and can be explained by differing datasets used for exposure and death rates and different exposure-response risk functions and counterfactual exposure. We cite the GBD 2019 data downloaded from the Institute for Health Metrics and Evaluation (IHME) center at the University of Washington (http://ghdx.healthdata.org) and we also cite the associated paper (Abbafati et al. 2020) in the reference list.